

# Evaluation of outbreak response immunization in the control of pertussis using agent-based modeling

Alexander Doroshenko[1], Weicheng Qian[2] and Nathaniel D. Osgood[2]

[1] Faculty of Medicine and Dentistry, Department of Medicine, Division of Preventive Medicine, University of Alberta, Edmonton, Alberta, Canada
[2] Department of Computer Science, University of Saskatchewan, Saskatoon, Saskatchewan, Canada

Corresponding author
Alexander Doroshenko,
adoroshe@ualberta.ca

## ABSTRACT

**Background**. Pertussis control remains a challenge due to recently observed effects of waning immunity to acellular vaccine and suboptimal vaccine coverage. Multiple outbreaks have been reported in different ages worldwide. For certain outbreaks, public health authorities can launch an outbreak response immunization (ORI) campaign to control pertussis spread. We investigated effects of an outbreak response immunization targeting young adolescents in averting pertussis cases.

**Methods**. We developed an agent-based model for pertussis transmission representing disease mechanism, waning immunity, vaccination schedule and pathogen transmission in a spatially-explicit 500,000-person contact network representing a typical Canadian Public Health district. Parameters were derived from literature and calibration. We used published cumulative incidence and dose-specific vaccine coverage to calibrate the model's epidemiological curves. We endogenized outbreak response by defining thresholds to trigger simulated immunization campaigns in the 10–14 age group offering 80% coverage. We ran paired simulations with and without outbreak response immunization and included those resulting in a single ORI within a 10-year span. We calculated the number of cases averted attributable to outbreak immunization campaign in all ages, in the 10–14 age group and in infants. The count of cases averted were tested using Mann–Whitney $U$ test to determine statistical significance. Numbers needed to vaccinate during immunization campaign to prevent a single case in respective age groups were derived from the model. We varied adult vaccine coverage, waning immunity parameters, immunization campaign eligibility and tested stronger vaccination boosting effect in sensitivity analyses.

**Results**. 189 qualified paired-runs were analyzed. On average, ORI was triggered every 26 years. On a per-run basis, there were an average of 124, 243 and 429 pertussis cases averted across all age groups within 1, 3 and 10 years of a campaign, respectively. During the same time periods, 53, 96, and 163 cases were averted in the 10–14 age group, and 6, 11, 20 in infants under 1 ($p < 0.001$, all groups). Numbers needed to vaccinate ranged from 49 to 221, from 130 to 519 and from 1,031 to 4,903 for all ages, the 10–14 age group and for infants, respectively. Most sensitivity analyses resulted in minimal impact on a number of cases averted.

**Discussion**. Our model generated 30 years of longitudinal data to evaluate effects of outbreak response immunization in a controlled study. Immunization campaign implemented as an outbreak response measure among adolescents may confer benefits across all ages accruing over a 10-year period. Our inference is dependent on having an

outbreak of significant magnitude affecting predominantly the selected age and achieving a comprehensive vaccine coverage during the campaign. Economic evaluations and comparisons with other control measures can add to conclusions generated by our work.

## INTRODUCTION

### Recent epidemiology of pertussis

In recent years, pertussis control has re-emerged as a prominent public health challenge, with multiple outbreaks observed worldwide (*Tan, Dalby & Forsyth, 2015*), and with some jurisdictions reporting the highest numbers of cases seen in decades (*California Department of Public Health, 2015*). In Canada, the last peak in pertussis activity was seen in the mid-1990s, after which incidence rates were gradually declining prior to a 2012 resurgence. This recent increase was driven by outbreaks in several provinces/territories (*Smith et al., 2014*). The national age-specific incidence rate remains highest among infants under 12 months of age, an age group also suffering the most hospitalizations and deaths. However, recently school-age children and younger adolescents have also borne a disproportionate burden, particularly during outbreaks. During the 2012 New Brunswick outbreak, the highest age-specific incidence rate fell in the 10–14 age group, which was twice as high as the incidence rate among infants (1,240 vs. 660 per 100,000, respectively) (*Office of the Chief Medical Officer of Health, 2014a*). Such increases in incidence rates among older children were reported from several US states (*California Department of Public Health, 2015*; *Wisconsin Department of Health Services, 2012*), suggesting a bimodal age distribution of cases in some jurisdictions. During three most recent Minnesota outbreaks, the proportion of pertussis cases among children 7–18 years old exceeded 60% (*Sanstead et al., 2015*).

The recent increase in pertussis activity is thought to be due to a combination of waning immunity from acellular pertussis vaccine and sub-optimal vaccine coverage. In the Ontario outbreak, cases were reported among unvaccinated individuals from a religious community and among vaccinated school-aged children (*Deeks et al., 2014*). In New Brunswick, 67% of cases in the 10–14 age group were up-to-date with their immunization (*Office of the Chief Medical Officer of Health, 2014a*). Several studies estimated the annual decline in protection after pertussis vaccination as ranging from 21% to 62% (*McGirr & Fisman, 2015*; *Klein et al., 2012*; *Tartof et al., 2013*). Vaccine-derived protection among individuals who were primed with the whole-cell pertussis vaccine is reported to be greater compared to individuals who received purely acellular formulations (*Sheridan et al., 2014*) Furthermore, natural disease confers even greater—but not life-long—protection (*Wendelboe et al., 2005*). Genetic mutations in the *Bordetella pertussis* bacterium and better detection and diagnosis have been suggested as other explanations for this recent pertussis trend (*Jackson & Rohani, 2014*).

Vaccination remains a cornerstone of public health measures to control pertussis. Improving immunization schedule adherence by raising awareness among public is the most commonly used intervention. The strategy of "cocooning" infants (vaccinating parents and other individuals in close contact with infants) has been advocated, with mixed reviews (*Rosenblum et al., 2014*; *Healy et al., 2015*). Immunizing pregnant women in the third trimester of pregnancy to prevent pertussis disease in infants too young to receive vaccination is recommended in the US (*Centers for Disease Control and Prevention, 2013*). Modifications of the immunization schedule (changing the age of vaccine administration or adding doses) have been discussed (*Libster & Edwards, 2012*). Developing new vaccines will offer the best long-term control strategy, however it is not likely to occur in the short term (*Meade, Plotkin & Locht, 2014*).

The ongoing occurrence of pertussis outbreaks presents a challenge to public health authorities which may necessitate supplementary control measures. In Canada, immunization of pregnant women is recommended only in outbreak situations (*Public Health Agency of Canada, 2014b*). Early contact tracing and chemoprophylaxis of contacts has been advanced as protective in control of school-based outbreaks (*Miguez Santiyan et al., 2015*). Outbreak response immunization (ORI) has been employed if a particular group is disproportionately affected and it is feasible to reach and vaccinate this group in a relatively short period of time (*Office of the Chief Medical Officer of Health, 2014a*). ORI is supplementary immunization given over and above the routine vaccination schedule, including to those who may be fully immunized or those who did not receive their scheduled vaccines. Potential benefits of ORI could accrue both in the short-term (terminating or limiting an ongoing outbreak) and long-term (preventing future outbreaks) and may extend to age groups other than the age group for whom ORI was targeted. The relevance of this intervention to all population (all other age groups) reflects the fact that interrupting transmission among mostly affected group may also decelerate transmission to other age groups with whom these individuals come in contact. Furthermore, it is very important to see whether stopping outbreak in the ORI target group can have a material effect on a particularly vulnerable population—babies under 1 year of age. ORI may also blunt natural boosting from circulating sub-clinical infections. The cost of such immunization campaigns, including emergency response infrastructure, cost of vaccines and their delivery is high, and often not included in routine immunization programs budgets. Evaluation of such immunization campaigns is limited and the need for pertussis outbreak response research has been advocated (*World Health Organization, 2009*). The objective of our study was to investigate the effect of outbreak response immunization (ORI) among adolescents as an emergency public health intervention in light of a recent re-emergence of pertussis outbreaks.

## Modeling approaches for pertussis

Previous studies have used aggregate deterministic and stochastic models to understand pertussis epidemiology and transmission and effects of vaccinations. In *Grenfell & Anderson*'s *(1989)* deterministic model, reduction of pertussis cases attributed to vaccination was estimated by comparing pre-vaccine and vaccine eras. Hethcote's model, originally published in 1997, introduced differential levels of immunity and infectiousness
(*Hethcote, 1997*). Hethcote subsequently contributed multiple adaptations of this model (*Hethcote, Horby & McIntyre, 2004*; *Van Rie & Hethcote, 2004*). Hethcote's models were adapted by other authors to evaluate effects of delays in pertussis immunization, improving vaccine coverage (*Pesco et al., 2015*; *Pesco et al., 2013*) and effectiveness of a routine adolescent booster (*Fabricius et al., 2013*). *Wearing & Rohani (2009)* used modeling to estimate duration of pertussis immunity. *Gambhir et al. (2015)* used US surveillance data to fit the model to estimate epidemiological and vaccine-related parameters responsible for recent increases in pertussis activity and demonstrated difference in duration of protection conferred by acellular versus whole-cell vaccine. *Sanstead et al. (2015)* developed an agent-based model to characterize pertussis outbreaks in Minnesota.

### Rationale for ABM

Impacts of interventions such as ORI cannot be summarized directly by collecting surveillance data because of the lack of controls (absent ORI intervention for the same outbreak). By contrast, such features and the complex interplay of waning immunity, network-mediated transmission, falling vaccination coverage, immunity boosting effects of exposure, and ORI and routine vaccination schedules make this investigation well-suited to agent-based simulation modeling (*Mabry et al., 2010*). Such models can be used to systematically evaluate health outcomes during pertussis outbreaks in an otherwise identical context in the presence and absence of ORI. An agent-based modeling approach was selected here due to several characteristics of the system involved, including—but not limited to—the important role of individuals' connections, the spatially clustered character of outbreaks, the need for a finer-grained representation of both age and waning immunity, and the need for a longitudinal lens to understand the impact of individual vaccination compliance on vulnerability and to calibrate vaccination coverage data. An agent-based approach was further valuable for representing certain ORI intervention scenarios, particularly the restriction of administration of ORI to those with particular classes of vaccination histories.

### Methods

We developed an agent-based simulation model (ABM) and estimated the age-specific effects of the pertussis ORI campaign in the 10–14 age group in simulated outbreaks in terms of the number of cases averted over the short-, medium- and long-term (1, 3 and 10 years following ORI implementation) among young adolescents (ORI target group), infants under 1 year and individuals of all ages. This study was approved by the Health Research Ethics Board at the University of Alberta, study ID Pro00050642.

### Model structure and agents characteristics

The essential structure of the agent-base model is shown in Fig. 1A. Agents representing individual persons were associated with both fixed attributes and evolving states. Fixed attributes included a location (detailed below) and vaccination attitude, while evolving aspects of agent state included (continuous) age, count of vaccinations received and count of pertussis infections contracted. Statecharts were used to represent the natural history of infection, demographics and vaccination schedule.

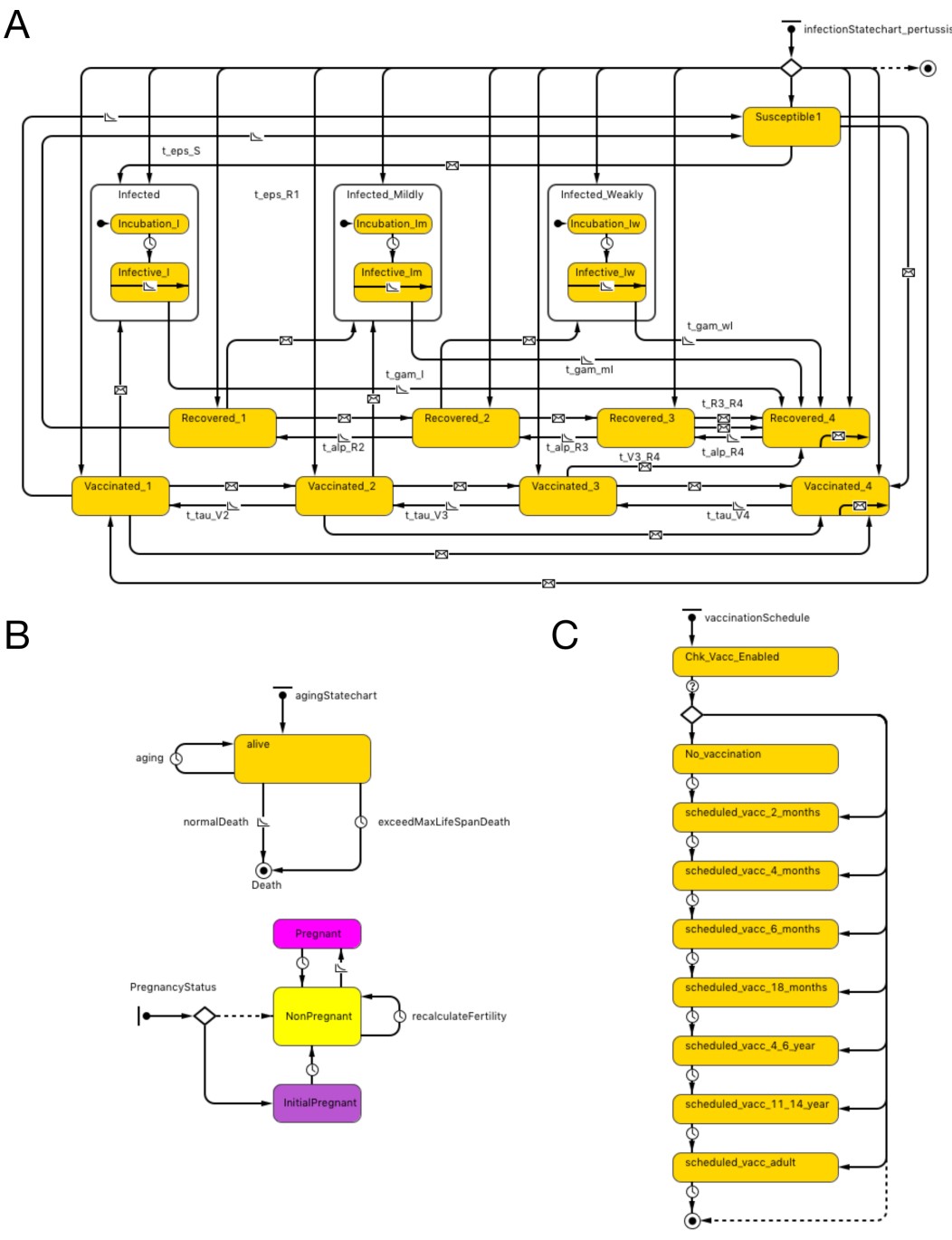

**Figure 1** **Model structure.** (A) Pertussis natural history statechart. (B) Demographic statechart. (C) Vaccination schedule statechart.

The natural history of infection statechart drew its structure from the characterization in Hethcote's widely published and adapted compartmental model (*Hethcote, 1997*). As described in previous contributions (albeit at a compartmental rather than individual level), this representation includes 3 levels of severity of infection (I-full, I-mild, I-weak), and four levels of vaccination- (V1–V4) and naturally-induced (R1–R4) immunity and

transitions between V, R and I states (with time and rate constants $\tau, \alpha$ and $\gamma$). Transitions were expressed by time-based and exposure-based formulations (e.g., 2 year increments to transition from higher to lower V states and 5 years to transition from higher to lower R states). In order to more realistically model the shapes of epidemiological curves over time to capture the outbreak dynamics that is critical to ORI triggering algorithms, we added incubation periods to the state charts, based on triangular distribution. While earlier adaptions of the Hethcote formulation typically assumed random mixing, in our model, individuals are importantly exposed to pathogens over a contact network (see below). In both cases, infection transmission is only possible to individuals who are susceptible (S state) or in the lower two levels of vaccine- and naturally- induced immunity. Vaccination and pathogen exposure boosts the level of both vaccine- and naturally induced immunity; waning of immunity decreases such immunity over time. While in an infected state, individuals in our model expose network connections, chosen with uniform probability, to pathogen at an age-specific contact rate.

Demographic statecharts reflected a possible lifespan varying from 0 to 100 years and illustrated individual mortality and (for female agents) fertility, with both being characterized using age-specific hazard rates derived from Hethcote model. We used the 2004 Canadian population pyramid to initialize the simulation (Fig. 1B).

The vaccination schedule statechart is modeled on North American vaccination regimes. It characterizes possible vaccination episodes at ages 2, 4, 6 and 18 months, and 4–6 and 11–14 years (dose 6 is given at 11–12 years of age in the US and in grades 7–9 in Canada), with one adult (18 years or older) vaccination also depicted. The range of ages when an adult dose is administered in our model is between 18 and 35. This reflects the fact that parents of young children may receive such a dose as a part of "cocooning" strategy. Alternatively, individuals presenting to emergency departments with tetanus-prone wounds may also receive a pertussis-containing vaccine. At each such age juncture, a person has a vaccine attitude- (acceptor, rejector or hesitant) (Tables S1A–S1B) and age- dictated probability of securing a vaccination encounter; conditional on such an encounter, a vaccine is delivered. In our model, we built a function to offer a catch-up vaccine if the previous dose is missed up to the minimal age of the next dose (e.g., if 5th dose is missed, an individual can have a chance to receive a catch-up dose up to the age of 11 years). The probability of offering catch-up immunization is 90%, but this is conditional on an individual having a further encounter with a vaccine provider, which in turn is dependent upon dose-specific vaccine coverage as well as contingent on personal vaccine attitude (e.g., a vaccine rejecter will not have a catch-up vaccine offered). Each occurrence of vaccination is associated with a fixed chance of vaccine failure (Fig. 1C).

## Network and spatial context

To capture the spatially clustered nature of outbreaks, agents were distributed throughout a stylized geographical area. The agent population was divided into a low density periphery (constituting 29% of the population but 89% of the area), and a central region of 20-fold higher population density (holding the balance of the population and occupying the remaining area).

**Table 1  Model's configuration and key parameters.**

| Parameter category | Parameter name | Description | Value |
|---|---|---|---|
| Demographics | Population size, (Persons) | Population size at the model's initialization | 500,000 |
| Disease mechanism | Incubation period, (Days) | Incubation period corresponding to different I states | Follows triangular distribution (min = {7, 10, 14}, mode = {10, 14, 21}, max = {14, 21, 42}) |
| | Mean waning time between R states, (Years) | | 5 |
| | Mean waning time between V states (whole-cell vaccine), (Years) | Determine vaccine-derived and natural disease-derived waning immunity | 2 |
| | Mean waning time between V states (acellular vaccine), (Years) | | 2 |
| Disease propagation | Exogenous infection Rate, (1/Day) | Represents imported infections | 5 |
| Network characteristics | Connection range, (Length) | Control mixing patterns and cumulative incidence and shape of epidemiological curve over time generated by the model | {Preferential = 55, Normal = 5} |
| | Base contact rate, (1/Day) | | {Preferential = 20, Ordinary = 3} |
| | Preferential mixing age, (Years) | | {from = 0, to = 16, years} |
| | Base population density, (persons/length$^2$) | | 0.002 |
| | Central-outer density ratio, (Unit) | | 20 |
| Vaccine coverage | Initial distribution of vaccination attitude in population, % | Determine vaccine coverage generated by the model | {Vaccine Acceptor (HA) = 50, Vaccine Hesitant (VH) = 40, Vaccine Rejector (VR) = 10} |

Based on their spatial location, agents were placed in a quasi-static assortive network in which a pair of agents was connected only if they lay within a specified distance threshold of each other. The model used different distance thresholds governing whether a given pair of persons was connected according to the age group of pair members. Specifically, while most pairs were connected only if they were within a certain range of each other, if both members of the pair were between 0 and 16 years old (inclusive), an 11-fold larger connection threshold was used guided by calibration.

## Parameterization

We configured our model using key parameters given in Table 1. Disease mechanism parameters pertaining to transitions between various $V, R$ and $I$ states were as described in the *Hethcote (1997)* model. A primary vaccine failure probability described in the literature (*McGirr & Fisman, 2015*), and incubation periods' range reflecting literature values (*Centers for Disease Control and Prevention, 2015*) were incorporated. Vaccine coverage was generated by the model. To simulate the dynamics of vaccine coverage, we classified all individuals into three groups: those who accept, reject and are hesitant to receive vaccination. For each of these groups, we assigned vaccination probabilities. By adding network characteristic parameters and an exogenous infection rate, we generated real-time epidemiological curves.

### Outbreaks and ORI triggers

We developed an automated algorithm for triggering ORI (Figs. S1A–S1C). The incidence rate of each month for the each age group was assigned a trichotomous S (sub-outbreak)

tag (S−, S, S+). S and S+ states would require exceedance of the 60-month moving average (excluding designated outbreaks) by 2 and 3 standard deviations, respectively and, additionally, exceedance of a specified monthly age-specific incidence rates (40 and 60 per 100,000, respectively); the latter being derived by examining surveillance and outbreak reports (*California Department of Public Health, 2015*; *Office of the Chief Medical Officer of Health, 2014a*; *Wisconsin Department of Health Services, 2012*; *Minnesota Department of Health Pertussis, 2012*) and further by calibration. An outbreak was defined as occurring if there were at least two consecutive months in the S state while ORI was triggered only in a setting of three consecutive months in S states or two consecutive months in S+ states. We computed the rates of simulated outbreaks and ORI occurrence by dividing the number of outbreaks and ORI interventions within a specified age group across all realizations by the product of a total number of realizations in a given experiment and the 30 years in each run (i.e., model run-years). For this study, we only triggered ORIs in the 10–14 age group. Reciprocals of the outbreak and ORI rates represent the mean period between outbreaks within a given age group and *time between ORIs*, respectively. The latter was used to evaluate sensitivity of our model to triggering ORIs. ORI implementation was modeled as achieving 80% vaccination coverage for all individuals aged 10–14 at the time of ORI administration.

## Calibration and validation

To better capture epidemiological trends of pertussis, we employed both quantitative and phenomenological approaches to validate and calibrate outputs from our model. More specifically, for the latter, we used the pattern-oriented modeling (POM) technique (*Grimm et al., 2005*; *Topping, Hoye & Olesen, 2010*; *Railsback & Grimm, 2012*) to optimize our model structure to ensure that generated epidemiological curves are realistic. We used this bottom-up strategy (*Railsback & Grimm, 2012*) to model the highly complex system to improve our model robustness in a situation when interplay of multiple factors governing pertussis transmission, outbreak propagation and vaccination dynamic are less predictable and the understanding of the characteristics of the causal pathways involved remains incomplete. We extensively varied most of our network parameters (connection range, contact rate, exogenous infections rate) and studied resultant patterns by visual inspection of baseline endemic activity, age-specific peaks of outbreaks and intervals between outbreaks using a graphical-user interface. We identified obviously unrealistic outputs and gradually converged towards a well-fitting combination of parameters.

Quantitatively, we adjusted model parameters to better match empirical cumulative incidence and dose-specific vaccine coverage. We calibrated the model to bring the 30-year cumulative incidence generated by the model in line with surveillance reports from two public health jurisdictions of similar population size in Alberta (Figs. 2A and S2) (*Alberta Health, 2015*). Comparability was defined as no more than 10% deviation between model-generated average cumulative incidence and that from two reference public health districts. Age-specific incidence rates were checked during calibration as described above to ensure that age groups with the highest burden of disease in our model were comparable to those of reference populations.

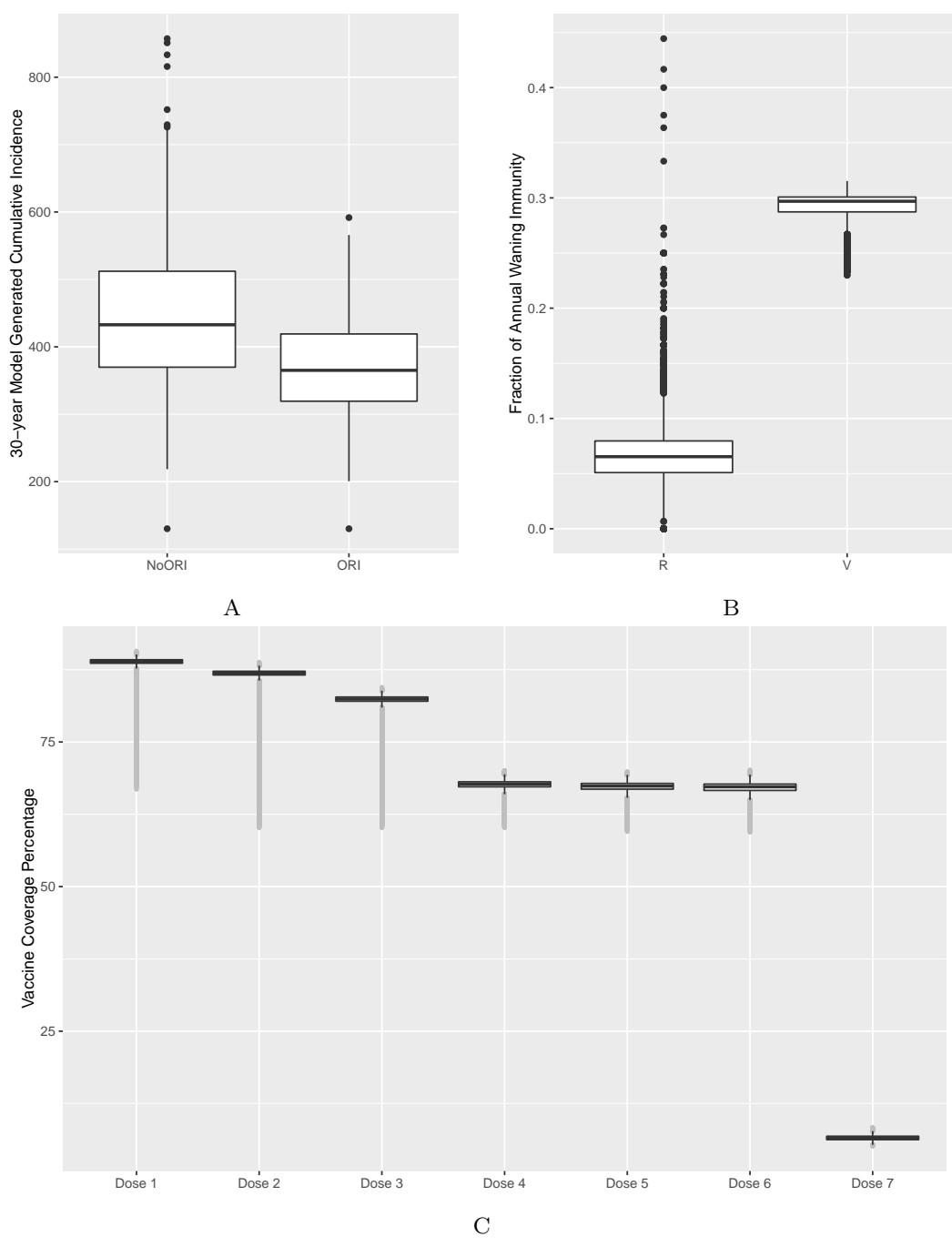

**Figure 2  Model's output validation and calibration.** (A) 30-year model-generated cumulative incidence. (B) Vaccine- and natural disease-derived waning immunity fractions. (C) Vaccine coverage by dose (doses 1–7). Model-generated outputs depicted in (A), (B) and (C) are compared to 30-year cumulative incidence derived from surveillance data from two Alberta jurisdictions, waning immunity values described in the literature and dose-specific vaccine coverage derived from Canadian data sources respectively.

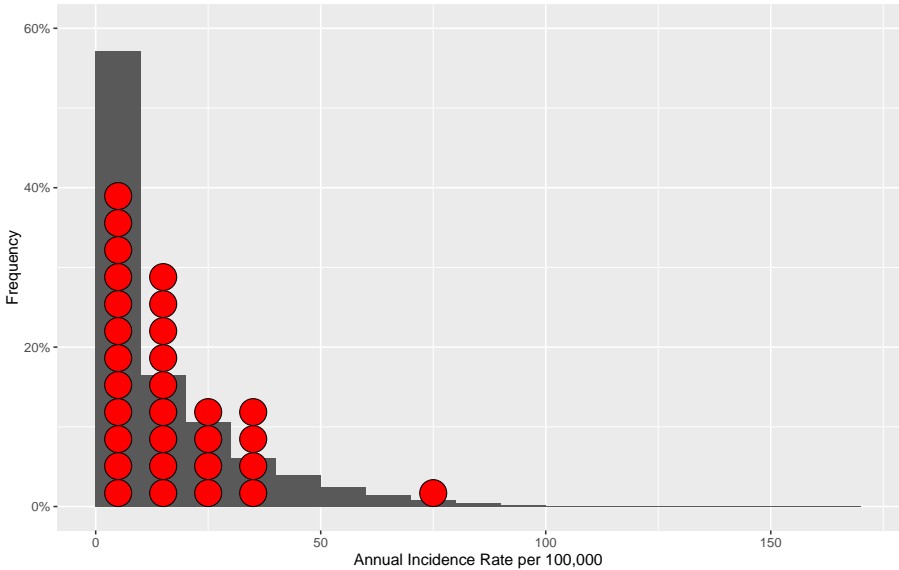

**Figure 3** **Distribution of the model-generated annual incidence rates in relation to empirical data.**
Bars represent frequencies corresponding to a particular annual incidence rate for the model-generated
data. Red circles represent reference populations' (Alberta Central and South Zones) annual incidence
rates. The total number of data points (annual incidence rates) based on which frequencies are computed
for model-generated data is 10,200.

We validated waning immunity outputs with data derived from literature. We defined
vaccine-derived and natural infection-derived waning immunity in the context of ABM
logic as transition from the (protected) V3 (or R3) state to (unprotected) V2 (or R2) for
each year. We generated a model output for waning immunity and illustrated it for the
4–16 age group (Fig. 2B, Figs. S3, S4A and S4B).

We calibrated vaccine coverage (as defined by receipt of all eligible doses) by varying
proportions of individuals by their vaccination attitudes and vaccination probabilities
assigned to their respective vaccination attitudes. The calibrated model was compared
against vaccine coverage statistics for doses 1–4 at age 2 published by *Alberta Health (2015)*
in the same public health jurisdictions used for cumulative incidence. Reference values
for doses 5, 6 and 7 vaccine coverage were obtained from other Canadian sources (*Public
Health Agency of Canada, 2015*; *Public Health Ontario , 2014*) given the reduced certainty
for these doses in Alberta (Fig. 2C).

To further test robustness of our model, we examined distribution of model-generated
annual incidence rates versus annual incidence rates distribution derived from empirical
surveillance data from two Alberta jurisdictions (Fig. 3) (*Alberta Health, 2015*). We tested
differences between these two data distributions by Mann–Whitney $U$-test ($p$-value =
0.1068) and by Kolmogorov–Smirnov test ($p$-value = 0.1075) with results indicating
reasonable representation of simulated data versus empirical data without evidence of
overfitting.

## Sensitivity analyses

As vaccine coverage for adult dose 7 may have the greatest uncertainty due to the lack of a fixed delivery age and underreporting of vaccination implemented as part of a cocooning strategy, we ran a sensitivity analysis increasing vaccine coverage for dose 7 by 20% (sensitivity analysis A). We further investigated the impact of waning immunity with a sensitivity analysis that reduced annual waning immunity by increasing transition time between V states ($\tau$) by 50% (from 2 to 3 years) among individuals born before 1997, representing the receipt of whole-cell vaccine (sensitivity analysis B).

Additionally, we performed sensitivity analyses by reducing naturally-derived waning immunity by doubling transition time between R states ($\alpha$) from 5 to 10 years (sensitivity analysis C) and by imposing eligibility restriction to receive ORI vaccination to only those who did not receive a regular vaccine within last 6 months (sensitivity analysis D).

We also performed sensitivity analysis by including a scenario positing a more rapid boosting effect from administration of vaccination (sensitivity analysis E). In this scenario, booster immunization for an individual in lower V states would lead immediately to a V4 state (full protection), without the multi-dose transition through V2 and V3 states as in the main experiment. Finally, we conducted a multi-way sensitivity analysis (sensitivity analysis F) where we considered the effects of simultaneous changes in ($\tau$) and ($\alpha$).

## Simulation setup and statistical analysis

An open model population of initial size 500,000 was simulated in continuous time using AnyLogic 7 software. We run multiple paired simulations (using identical random seeds) with and without enabling the automated ORI module for 33 years. The first three years of simulation were designated as a "burn-in period" and discarded, resulting in a 30-year period of observations per simulation run. To yield meaningful results, statistical analysis was only performed on "qualified" pairs of simulations, as judged by the following criteria:

(i) At least one ORI was triggered within a simulation run.
(ii) At least a 10-year post-ORI observation period was available.
(iii) There was no second ORI triggered within a 10-year observation period.

Furthermore, qualified simulations meeting above criteria had to exhibit a cumulative incidence rate comparable to two Alberta jurisdictions as described above.

We ran simulations on a high-power computer cluster at the University of Saskatchewan for 200 node-hours resulting in 334 pairs. Analyses were performed on data generated by the model. An illustration of differences between with-ORI and no-ORI case counts in a single paired simulation run for all ages before and after a triggered ORI is shown in Fig. S5. For each qualified pair, we calculated a number of cases averted within 1, 3 and 10 years after ORI for three age groups: all ages, 10–14 (the ORI target age group) and infants under 1 year of age (the most vulnerable group). The differences in the count of cases between the ORI and no-ORI groups for a given qualified simulation pair were tested using the one-way Mann–Whitney $U$ test to determine statistical significance. We calculated a number needed to vaccinate during ORI (NNV-ORI) to prevent a single case directly from the model by dividing a number of vaccinations delivered during an ORI by a number of

cases averted in a respective age-group. This quantity will vary significantly in the context of different assumptions regarding the population size and population immunity and therefore only applies to our model. Given multiple simulations, we reported minimum and maximum NNV values.

## RESULTS

189 qualified paired-runs met the inclusion criteria and were analyzed in the main experiment. Characteristics and summary statistics of model-generated data are shown in Table 2. In the main experiment, the outbreak rate for all ages was 0.315, indicating that outbreaks in all-ages group were occurring approximately every 3 years in the model. The respective frequencies of outbreak occurrence among adolescents 10–14 years of age and infants under 1 year of age were approximately once in 7 and 13 years. Peak annual and monthly incidence rates for all ages recorded in the main experiment across all realizations were 170 and 26 per 100,000, respectively. Similarly, while the overall outbreak rates were lower in the 10–14 year category and among infants when compared with those outbreak classifications considering all age groups, peak annual and monthly incidence rates for children 10–14 years were 931 and 181 per 100,000, respectively, and for infants under 1 year of age, 725 and 207 per 100,000. On average, an ORI campaign was triggered every 26 years. There were few variations in the rates of outbreaks' occurrence and ORI triggering between the main experiment and all sensitivity analysis scenarios other than sensitivity analysis involving more rapid boosting from vaccination (sensitivity analysis E), which demonstrated significantly lower outbreak and ORI frequency. The scenario in sensitivity analysis E was also the only situation where comparability criteria with empirical data were not met. In the course of pattern-oriented modeling (POM) validation, we observed that reducing the exogenous infection rate resulted in a lower background incidence rate, punctuated by more pronounced outbreaks for a given cumulative incidence.

Vaccine–induced and natural disease-derived waning immunity rates in our model were calculated to be 29% and 6.5% per year, respectively, in line with values reported from the literature (*McGirr & Fisman, 2015*; *Wendelboe et al., 2005*). Our model generated the following vaccine coverage for doses 1–7: 89%, 87%, 82%, 68%, 67%, 67% and 7%, respectively (Fig. 2C, Figs. S3, S4A and S4B).

In the main experiment, on a per-run basis, there were an average of 124, 243 and 429 pertussis cases averted across all age groups within 1, 3 and 10 years of a campaign, respectively. During the same time periods, 53, 96, and 163 cases were averted in the 10–14 age group, and 6, 11, 20 in infants under 1 ($p < 0.00001$ for all groups of comparisons of counts of cases in ORI versus no-ORI simulations, one-way Mann Whitney $U$ test). NNV-ORI ranged from 49 to 221, from 130 to 519 and from 1,031 to 4,903 for all ages, the 10–14 age group and for infants, respectively (Table 3). Boxplots for the number of cases averted for durations following ORI are depicted in Fig. 4, with each data point being associated with a particular realization. Over a 10-year period, there was a gradual accrual of cases averted across all studied age groups; however, the accrual rate exhibited diminishing gains in years 5 through 10.

**Table 2  Characteristics and summary statistics of the main experiment and sensitivity analyses.**

| | Description of parameter(s) alterations | ORI rate in 10–14 age group[a] per model run-years | Outbreak rate in all ages[b] per model run-years | Outbreak rate in under 1[b] per model run-years | Outbreak rate in 10–14 age group[b] per model run-years | Comparability with benchmark cumulative incidence[c] |
|---|---|---|---|---|---|---|
| Main experiment | Reference | 0.038 | 0.315 | 0.075 | 0.129 | Yes |
| Sensitivity analysis A | Increase vaccine coverage for dose 7 by 20% | 0.037 | 0.315 | 0.080 | 0.127 | Yes |
| Sensitivity analysis B | Increase value of ($\tau$) to 3 among those born before 1997 | 0.039 | 0.323 | 0.075 | 0.127 | Yes |
| Sensitivity analysis C | Increase value of ($\alpha$) to 10 | 0.040 | 0.341 | 0.082 | 0.135 | Yes |
| Sensitivity analysis D | Restrict ORI eligibility to those who did not receive vaccine within last 6 months | 0.036 | 0.324 | 0.078 | 0.129 | Yes |
| Sensitivity analysis E | Implement stronger vaccine boosting effect | 0.005 | 0.047 | 0.013 | 0.021 | No |
| Sensitivity analysis F | Multi-way sensitivity analysis B and C combined | 0.025 | 0.325 | 0.083 | 0.129 | Yes |

Notes.

[a] ORI rate is computed by dividing the number of triggered ORIs by the product of a total number of simulation runs in a given experiment and 30 years in each run. Reciprocal of the ORI rate represents mean time between occurrences of triggering ORIs; for example, the rate of 0.038 per model run-years in the main experiment indicates that ORI in the 10–14 age group was triggered every 26 years in the model ($1/0.038 = 26.3$).

[b] Outbreak rate is computed by dividing the number of outbreaks within a specified age group (or when judged with respect to all age groups) by the product of a total number of simulation runs in a given experiment and 30 years in each run. Reciprocal of the outbreak rate represents the mean period between outbreaks occurring within a given age group; for example, the rate of 0.315 per model run-years in the main experiment indicates that outbreaks in all age groups were occurring every 3 years in the model ($1/0.315 = 3.17$).

[c] Comparability with benchmark cumulative incidence was defined as model-generated 30-years cumulative incidence rate falling within 10% of the average empirical cumulative incidence rate derived from 15 years of observations in two jurisdictions in Alberta (15 years of observations were up-scaled to derive 30-year cumulative incidence).

**Table 3  Number of pertussis cases averted and numbers needed to vaccinate by time periods after the outbreak-response immunization campaign and by age groups: modeling-generated results.**

| Age groups | Post-outbreak-response immunization period, years | Average number of cases averted[*] | Minimum number needed to vaccinate[a] | Maximum number needed to vaccinate[a] |
|---|---|---|---|---|
| All ages | 1 | 124 | 171 | 221 |
| All ages | 3 | 243 | 87 | 112 |
| All ages | 10 | 429 | 49 | 64 |
| Under 1 year | 1 | 6 | 3,784 | 4,903 |
| Under 1 year | 3 | 11 | 1,834 | 2,377 |
| Under 1 year | 10 | 20 | 1,031 | 1,336 |
| 10–14 years old | 1 | 53 | 400 | 519 |
| 10–14 years old | 3 | 96 | 220 | 285 |
| 10–14 years old | 10 | 163 | 130 | 168 |

Notes.

[*] $p < 0.00001$ for all groups of comparisons of counts of cases in outbreak-response immunization (ORI) versus no-ORI simulations, one-way Mann Whitney $U$ test.

[a] Number needed to vaccinate (NNV) was calculated directly from the model by dividing a number of vaccinations delivered during the ORI by a number of cases averted in a respective age group. NNV only applies to a current model and for a given population size.

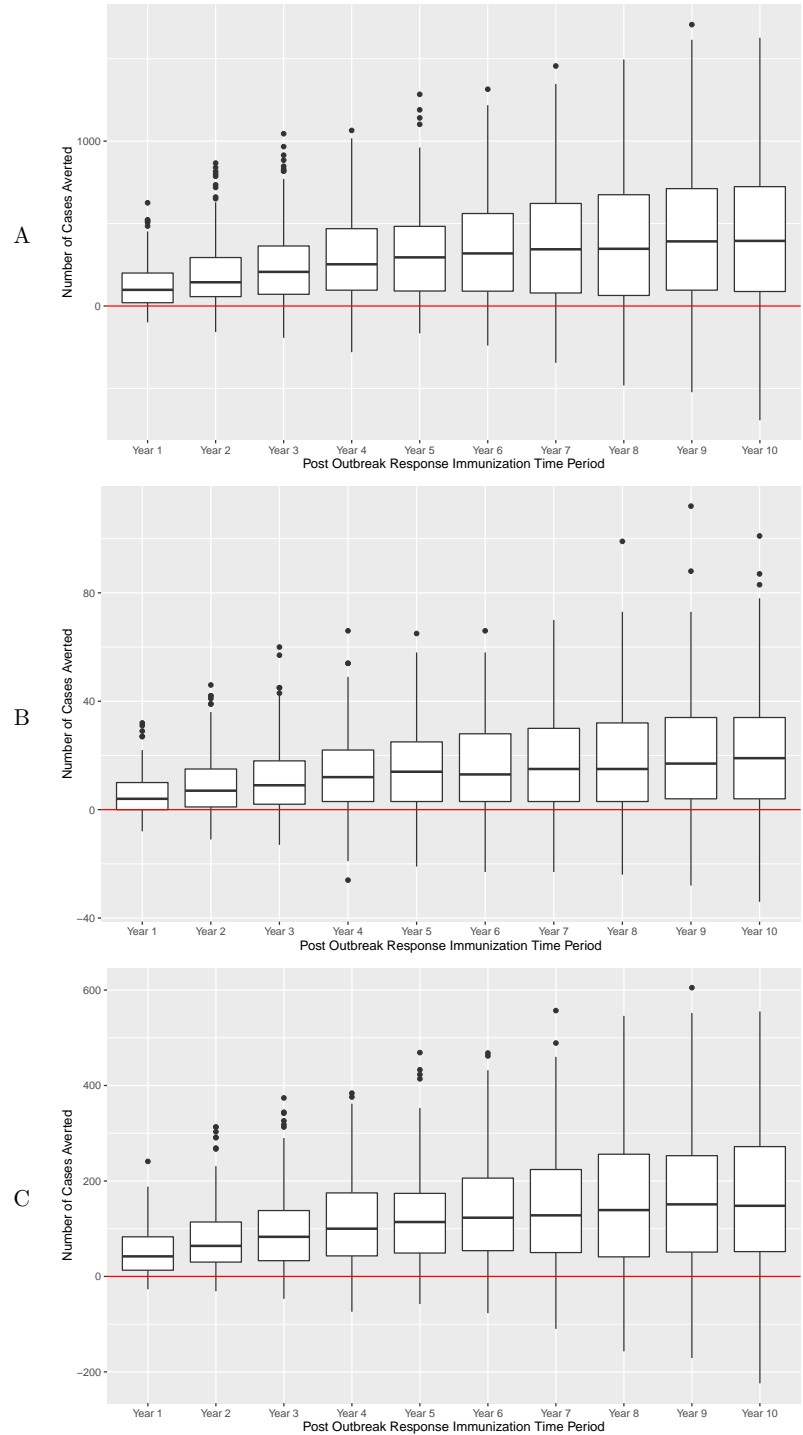

**Figure 4  Number of pertussis cases averted over a 10-year period after implementation of ORI campaign, main experiment.** (A) All ages group. (B) Infants under 1 year of age. (C) Adolescents 10–14 years of age. Results of sensitivity analyses are summarized in Table 4.

**Table 4  Number of pertussis cases averted: summary of sensitivity analyses.**

| Age group | Post-ORI period, years | Number of cases averted[a] by experiment | | | | | | |
|---|---|---|---|---|---|---|---|---|
| | | Main experiment[*, b] | Sensitivity analysis A[*, b] | Sensitivity analysis B[*, b] | Sensitivity analysis C[*, b] | Sensitivity analysis D[*, b] | Sensitivity analysis E[*] | Sensitivity analysis F[*, b] |
| All ages | 1 | 124 | 119 | 112 | 148 | 108 | 41 | 130 |
| All ages | 3 | 243 | 256 | 228 | 262 | 241 | 93 | 229 |
| All ages | 10 | 429 | 410 | 409 | 422 | 429 | 148 | 378 |
| Under 1 | 1 | 6 | 5 | 5 | 7 | 4 | 1[**] | 6 |
| Under 1 | 3 | 11 | 12 | 10 | 13 | 10 | 4 | 10 |
| Under 1 | 10 | 20 | 20 | 20 | 21 | 21 | 6 | 18 |
| Age 10–14 | 1 | 53 | 51 | 48 | 61 | 47 | 19 | 54 |
| Age 10–14 | 3 | 96 | 102 | 92 | 104 | 96 | 40 | 90 |
| Age 10–14 | 10 | 163 | 156 | 156 | 161 | 162 | 60 | 143 |

Notes.

[a] The number of cases averted was determined by subtracting count of pertussis cases in the simulation ORI arm from the no-ORI arm within the same experiment for a given age group and a given post-ORI period (count of cases expected to be lower in an ORI arm if cases are averted) and expressed as an average on a per-run basis.

[*] $p < 0.00001$ for all groups of comparisons of counts of cases in outbreak-response immunization (ORI) versus no-ORI simulations, one-way Mann Whitney $U$ test ($p$-values apply to the entire column).

[b] Denotes comparability to benchmark 30-year cumulative incidence rate (applies to the entire column).

[**] $p = 0.12$.

Prolonging the duration of vaccine-induced immunity among those who received whole-cell vaccine had minimal impact on overall waning immunity and number of cases averted by ORI. Prolonging the duration of natural disease-derived immunity resulted in such an annual waning immunity decreasing to 2% but had minimal impact on the number of cases averted. Equally, multi-way sensitivity analysis of combining both the effect of prolongation of whole-cell vaccine-induced and natural disease-derived immunity had little impact on the number of cases averted. Increasing vaccine coverage for dose 7–26% and restricting eligibility to receive vaccination during the ORI campaign to those who did not receive pertussis vaccine within 6 months had also minimal impact on cases averted relative to the baseline. However positing a significantly stronger boosting effect from vaccination (bypassing V2 and V3 states in the state chart in Fig. 1A) resulted in significantly fewer cases averted by ORI (148 cases averted in sensitivity analysis E versus 429 in the main experiment for all-ages group over 10 years). Also, the difference between ORI and no-ORI arms was not statistically significant for infants under 1 years of age at 1 year post ORI campaign ($p$-value = 0.12) in sensitivity analysis E. Interested readers can refer to the supplemental material for additional results of sensitivity analyses (Tables S2A–S2F and Figs. S6A–S6F).

## DISCUSSION

Our ABM successfully generated 30 years of longitudinal data to evaluate the effects of supplemental ORI in a controlled study. For this purpose, we expanded mechanisms widely adopted from a previously published pertussis compartmental model by developing a spatially-localized 500,000-person contact network representing a typical small-to-moderate size Canadian public health district, and also supplemented such elements

with novel mechanisms to dynamically recognize outbreaks suitable for ORI, and trigger resulting immunization campaigns.

Modeling is used to enhance fundamental understanding of pertussis characteristics and transmission and to more pragmatically evaluate impacts of interventions (e.g., adolescent or adult routine vaccination or cocooning strategy). While the latter is often a subject of recent enquiries, our model, to our knowledge, is the first to represent and evaluate the effects of pertussis ORI. Such an ORI-specific evaluation is an important contribution to our understanding of outbreaks dynamics, as the force of infection of the sort of focused, large scale outbreak needed to motivate ORI may generate different transmission patterns which cannot be seen in the non-outbreak settings, and because ORI can re-shape both short- and long-term transmission dynamics either for the benefit or possibly to a detriment. The large scale outbreak itself may exhaust the pool of susceptibles and consequently yield a decrease in the number of cases in post-outbreak years, and lower incidence can lead to diminished natural boosting. For example, annual pertussis incidence rates were at historically low levels in 2 years following a large scale outbreak in New Brunswick in 2012 (187, 0.5 and 1.2 per 100,000 in 2012, 2013 and 2014, respectively) with a smaller outbreak reported in the third year (*Office of the Chief Medical Officer of Health, 2013*; *Office of the Chief Medical Officer of Health, 2014b*). While this observation could be due to the effect of the outbreak itself, the contribution of the ORI (which was implemented in New Brunswick 2012 outbreak) is an important consideration.

We conclude that the effect of ORI is beneficial independently of the effect of the outbreak itself and leads to a net number of cases averted in all age groups, particularly in the short and medium term. While the objective of this model project focused on evaluation of the effects of ORI, the model also supported a set of interesting secondary observations. We found that reducing the exogenous infection rate resulted in a lower background incidence rate punctuated by more pronounced outbreaks. This may suggest that jurisdictions with lower migration may be more prone to larger scale but less frequent outbreaks, while jurisdiction with higher migration may exhibit more frequent outbreaks with lower peak incidence. No significant changes to our conclusions were observed from positing prolonged duration of natural disease-derived immunity, increasing adult vaccine-coverage or restricting vaccination eligibility during ORI. We observed no effect of altering the assumptions concerning waning immunity for those who received whole-cell vaccine, which may be due to the fact that our model ran prospectively into the future, with the number of individuals who had whole-cell vaccines progressively decreasing over time. However, our findings in a sensitivity analysis that positing a stronger boosting effect of vaccination implies a notably reduced burden of pertussis supports current thinking that insufficient duration of immunity contributes to the recent resurgence of pertussis outbreaks.

One of the considerations in modeling/reproducing outbreaks is that, while historical surveillance data plays an important role in defining whether an outbreak exist or not, the identification of an outbreak is often judgement-based, with similar magnitude of pertussis incidence determined to be an outbreak in a one jurisdiction, but not in others. We set outbreak and ORI thresholds in our model high, effectively excluding instances of "borderline" outbreaks where ORI is unlikely to ever be a consideration. As a result,
ORI in our simulations was triggered once every 26 years on average. This reflects the reality that ORI is not a commonplace intervention, particularly if disease is endemic. In our model, we implemented ORIs only to adolescents 10–14 years of age, reflecting recent outbreaks affecting this age group, who are largely fully immunized (and for whom immunization schedule adherence was not protective) and their accessibility to school-mediated campaigns; however, our model has the capability to test outbreak response in any age group. To ascertain whether ORI administered to young adolescents confer an indirect protection to other age groups via interruption of transmission, we specifically examined the effects of ORI administered to the adolescent age group on the number of cases averted among individuals of all ages and among infants, as protecting infants is one the main priorities for public health interventions. While we observed protective effect among adolescents and individuals of all ages, our study revealed that a protective effect to infants is modest, as suggested by high NNV generated by our model. These results are in the agreement with recent recommendations concluding that a booster dose in adolescence or adulthood had minimal impact on infant disease (*World Health Organization , 2014*); however, the latter recommendation was not specifically in the ORI context.

The main strength of our study is that we analysed longitudinal data generated by the model in a manner of a controlled study, thus allowing us to independently evaluate and quantify the effects of the ORI. As propagation of outbreaks depends on both intrinsic characteristics of individuals as well as transmission-permitting connections, which exist between these agents, including both characteristics in a single agent-based model allowed us to examine their interplay in outbreak occurrence. Our model included age-structure to model pertussis vaccination and incorporated vaccination attitudes into determination of vaccine coverage. Our model quantified both vaccine-induced and natural disease-derived waning immunity. Furthermore, we calibrated and validated the model by statistical comparison of the model-generated data and observed surveillance data as well as by utilizing pattern-oriented modeling. Our model could be adapted, with varying levels of ease, for different contexts and to investigate different types of research questions. Adaptation to investigate similar ORI phenomena in other jurisdictions would involve a circumscribed set of changes, including primarily changes to the vaccination statechart and associated probabilities (to represent local vaccination regimes), probabilities associated with vaccine attitude (reflecting differences in local attitudes towards vaccination), population sizes and population density, and potentially age-specific mixing assumptions. With a greater degree of modifications, and contingent on retaining current disease transmission logic, our model would also permit to investigate effects of other public health interventions ranging from altering vaccination schedules, evaluating effects of passive messaging to adhere to immunization schedules and adding vaccine doses in adults.

Our study has several limitations. We used disease mechanism parameters initially outlined in the Hethcote model. While conducting several sensitivity analyses involving key parameters, our experiments with different disease transmission logic were limited to enhancing boosting effect; a broader set of altered assumptions in this area may or may not yield different results for our research question. Recent study suggests that non-human primates vaccinated with acellular pertussis vaccine were protected from

severe symptoms, but not infection, and readily transmitted *Bordetella pertussis* to contacts (*Warfel, Zimmerman & Merkel, 2014*). In recent review of pertussis models, *Campbell, McCaw & McVernon (2015)* identified incomplete understanding relating infection and disease and lack of supporting data to derive parameters as common limitations of proposed pertussis models. While our calibration process helped ensure that our model output is realistic, we did not test variations in every single parameter given the multi-faceted nature of our model. Furthermore until further knowledge emerges to narrow down or alter parameters value, using the classic model structures on which our model is based would appear appropriate. We did not aim to examine and compare public health strategies other than ORI, and the need to pursue such research is strong. Economic evaluations can offer valuable additions to conclusions generated by our work.

## CONCLUSIONS

We developed an agent-based model to investigate effects of outbreak response immunization campaigns targeting young adolescents in averting pertussis cases. We concluded that such an immunization campaign confers benefits across all age groups accruing over a 10-year period. Our inference is dependent on having an outbreak of significant magnitude affecting predominantly the selected age and achieving a comprehensive age-specific coverage rate during the campaign. Our results demonstrated that while outbreak response may yield modest benefits for protecting infants, additional strategies to protect this vulnerable group are needed. Our experience indicates that ABM offers a promising methodology to evaluate other public health interventions used in pertussis control. We also identify the strong need for further research into application of modeling to further our understanding of pertussis epidemiology.

## ACKNOWLEDGEMENTS

Authors acknowledge contributions of public health departments in Alberta and New Brunswick in obtaining surveillance data.

### Funding

This work was supported by the Alberta Health grant and A. Doroshenko's start-up funds from the University of Alberta. The funders had no role in study design, data collection and analysis, decision to publish, or preparation of the manuscript.

### Grant Disclosures

The following grant information was disclosed by the authors:
The Alberta Health grant.
A. Doroshenko's start-up funds.

### Competing Interests

The authors declare there are no competing interests.

## Author Contributions

- Alexander Doroshenko conceived and originated the study, wrote the paper, led the research group, contributed to the study design, development of experiments, model configurations, running calibrations and validations, data analysis and interpretation as well as writing and editing the paper.
- Weicheng Qian programmed and ran model simulations, served as the primary model designer and data analysis lead, contributed to the study design, development of experiments, model configurations, running calibrations and validations, data analysis and interpretation as well as writing and editing the paper.
- Nathaniel D. Osgood supervised all aspects of model development, contributed to the study design, development of experiments, model configurations, running calibrations and validations, data analysis and interpretation as well as writing and editing the paper.

## Ethics

The following information was supplied relating to ethical approvals (i.e., approving body and any reference numbers):

This study was approved by the Health Research Ethics Board at the University of Alberta: Study ID Pro00050642.

## Data Availability

Qian, Weicheng; Doroshenko, Alexander; Osgood, Nathaniel (2016): Pertussis ABM model. figshare. https://dx.doi.org/10.6084/m9.figshare.3384106.v1.

## Supplemental Information

Supplemental information for this article can be found online at http://dx.doi.org/10.7717/peerj.2337#supplemental-information.

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
