# Peer review of "Evaluation of outbreak response immunization in the control of pertussis using agent-based modeling"

_PeerJ, doi:10.7717/peerj.2337_

## Round 0.1 · original submission · Major Revisions

After a proper observation, and after the assessment of the comments of the Reviewers, the manuscript entitled " Evaluation of outbreak response immunization in the control of pertussis using agent-based modeling" is not recommended for publication in its present form. The manuscript it is presented in a confusing way. Moreover, regarding the part of the models and the validation and calibration it needs more improvements.

The authors should respond to all the reviewers comments in their revision.

Reviewer 1 ·

Basic reporting

Not sufficient background on related modelling.
Limited Results. Poor Figure elements.

Experimental design

No Comments.

Validity of the findings

Lack of testing of model robustness.

Additional comments

Interesting paper. Manuscript is well written and clear to read. However, I think that there are some issues in the methods part and in the results sections that should be addressed.
At the moment I do not think it's ready for publication.
1. The authors give just a very brief rational for ABM, but they do not satisfactorily cover the review of the available modelling approaches for pertussis, in the form of ABM models or more canonical differential-equation models.
This is an important point because they could compare the results obtained with their model with other approaches.
2. The model validation and calibration is not very comprehensive. It is basically limited to matching the outcome incidence and vaccine coverage. However, it does not cover any model robustness (e.g. forecasting or performance on unseen data) nor internal model selection/parameterization. for instance there is no evaluation of rules' importance or extensive parameter shuffling. The sensitivity analysis should be extended as well.
3. The results section is too brief. Here, description of the population characteristics, of the important disease triggers, of different vaccine scenarios, etc, should have been provided thoroughly. also, comparisons with other models should have been given to understand how the model improves over the current state-of-the-art.

Reviewer 2 ·

Basic reporting

No Comments

Experimental design

No Comments

Validity of the findings

No Comments

Additional comments

Doroshenko et al., performed an investigation about the effects of an outbreak response immunization targeting young adolescents in adverting pertussis cases.

Comments:
- It is not clear where the performed the analysis? In Canada or worldwide?
- The sample is not clear. How is it composed? How is it selected? What is the source of the data? Please clarify better.
- It is not clear how the statistical analysis was performed.
- The paper is confusing and boring. Try to write better the discussion section making it more readable.

Reviewer 3 ·

Basic reporting

Doroshenko et al. in this manuscript evaluated the effects of of an aoutbreak response immunization targeting young adolescents in the control of pertussis using agent-based modeling.

Experimental design

Authors declare in the aim of the study that the study was directed to the population age 10-14 years but then along the manuscript emerged that results are relative to all population. Please, clarify.

The first part of the “methods” section should be moved to the “Introduction” section.
The model used for the study should be expolained in a more clear way to be understandable to normal readers. A schematic representation in a figure should be useful.

Validity of the findings

Results are referred to the population 10-14 age but the discussion of the results is related to the whole population: please verify and explain this contrast.

Additional comments

Comments to the authors

The manuscript is very interesting, Authors suggested an epidemiological model, agent-based model, for pertussis transmission evaluation that can represent a valid tool to foresee if in a population an outbreak of pertussis could happen. This prevision can be made in relationship with the changeable adherence to campaign of vaccination and to the vaccine coverage.

---

## Round 0.2 · Minor Revisions

The authors performed a good revision of this manuscirpt but they have to try to discuss the final remaining comment highlighted by Reviewer 1.

Reviewer 1 ·

Basic reporting

The authors have adequately addressed all my previous concerns. I would only ask an additional thing to be discussed in the paper. How could their modelling approach be generalized in a different context for reuse, testing hypotheses and forecasting? It would be useful because of recent outbreaks due to vaccinations drop in other countries (e.g. Spain, Italy recently).

Experimental design

No Comments

Validity of the findings

No Comments

Additional comments

No Comments

Reviewer 2 ·

Basic reporting

No Comments

Experimental design

No Comments

Validity of the findings

No Comments

Additional comments

All comments have been addressed.
the authors performed a very good revision of the paper.

---

## Round 0.3 · accepted · Accept

The authors improved the manuscript and I think that now it is ready to be published. Particularly the authors added the part related on how could their modelling approach be generalized in a different context for reuse. So, my suggestion for this manuscript is to Accept it !